# Sustainability by High–Speed Rail: The Reduction Mechanisms of Transportation Infrastructure on Haze Pollution

**Yu Chen [1], Yuandi Wang [1] and Ruifeng Hu [2,*]** 

[1]  Business School, Sichuan University, Chengdu 610064, Sichuan, China; scuchenyu@gmail.com (Y.C.);
    wangyuandi@scu.edu.cn (Y.W.)

[2]  School of Economics, Xihua University, Hongguang Avenue 9999, Chengdu 610039, Sichuan, China

*   Correspondence: huruifeng.scu@gmail.com

**Abstract:** Haze pollution impacts human health, as well as the competitive capabilities of firms and local economic development. Considerable attention has been given to the study of mechanisms for reducing haze pollution, but few studies have investigated the effect of city-to-city transportation infrastructures on environmental issues based on an institutional perspective. To address this research gap, this study advances our understanding by assessing the effect of city–to–city transportation on haze pollution caused by the operation of high-speed rail, which triggers the rapid flow of individuals and information, improves information transparency, as well as imposes institutional pressure on local governments and firms to reduce haze pollution. To further verify the underlying mechanisms, we tested the development of hard infrastructure (information communication technology) and soft infrastructure (market development level), which represent two conditions for which the mechanism is likely to be critical. We tested our hypotheses using a sample of 288 prefecture-level cities in China during the period from 2005 to 2016. The empirical results indicate that the operation of high-speed rail can reduce haze pollution by 17% on average.

**Keywords:** high-speed rail; transportation infrastructure; haze pollution; information transparency; institutional pressure

---

## 1. Introduction

Haze pollution has been the subject of increasing concern among both academics and practitioners alike, due to its serious consequences in the long term. It has been shown to have direct influences on the health and life spans of residents, the innovation capabilities of firms, and the economic development of local cities [1–5]. According to a World Health Organization report in 2016, almost 3 million deaths every year are related to air pollution, of which 94% of these deaths are due to respiratory infections, lung disease, and lung cancer because of the inhalation of particulate matter (PM). In addition, cities with severe air pollution are likely to experience a loss of talent and a decrease in regional vitality and creativity. Haze pollution, as one type of air pollution, is mainly caused by particulate matter and can be easily observed compared with other pollutants, such as $SO_2$, $NO_2$, or CO, because of its visibility.

Thus, identifying mechanisms for reducing haze pollution has become an urgent task. Previous studies have mainly focused on mechanisms that lead to stricter regulations, technological innovation, and infrastructure improvement [6–8]. Governments have implemented various incentives and punishments to encourage subordinate departments to protect the environment, such as tax exemptions, financial subsidies, penalties, and policies [9]. Technological innovation is also an important reduction mechanism of environmental pollution, as it can help firms to improve production processes as well as reduce emissions of toxic gases [6].

Infrastructure improvement, such as road intensity and transportation infrastructure, has been suggested as an important mechanism for reducing haze pollution; however, few studies have focused on the relationship between haze pollution and long-distance transportation across cities and provinces [8,10]. Previous studies initially highlighted the role of urban transportation, such as urban rail transit, bus rapid transit (BRT), and electric cars, in improving air quality in terms of the associated reduction of CO emissions and other pollution sources [10–12]. However, it is not yet known whether there is a relationship between high-speed railway and pollution emissions. Figure 1 depicts China's high-speed rail (HSR) network. Figure 2 shows the macro–level relationship between the development of high-speed rail and the pollution emissions, intuitively indicating that with the development of HSR, the pollution seems to have gradually decreased. Therefore, it is important to determine the relationship between high-speed railway and pollution emissions.

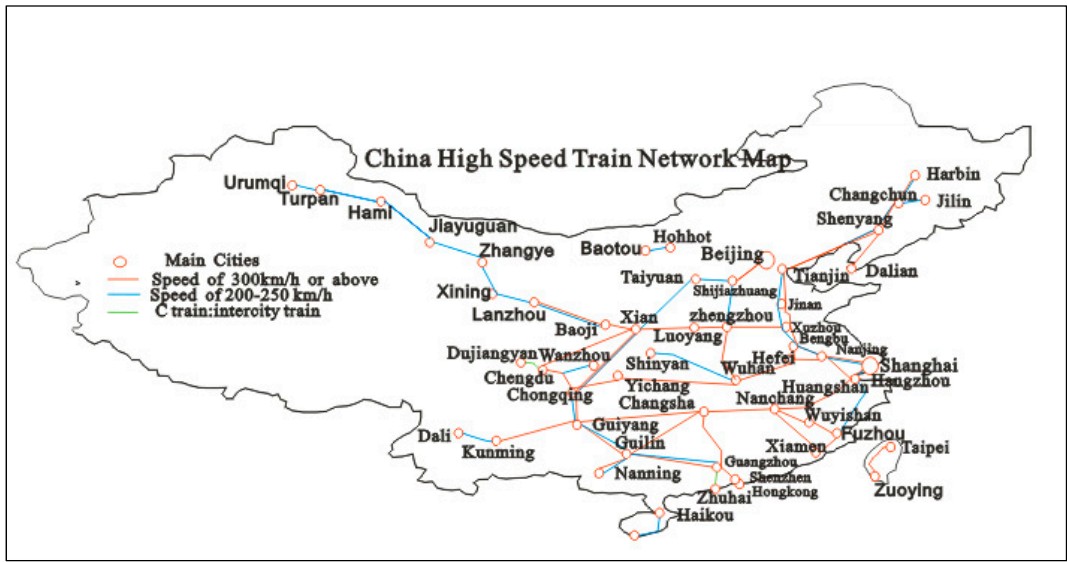

**Figure 1.** China's high-speed rail network map.

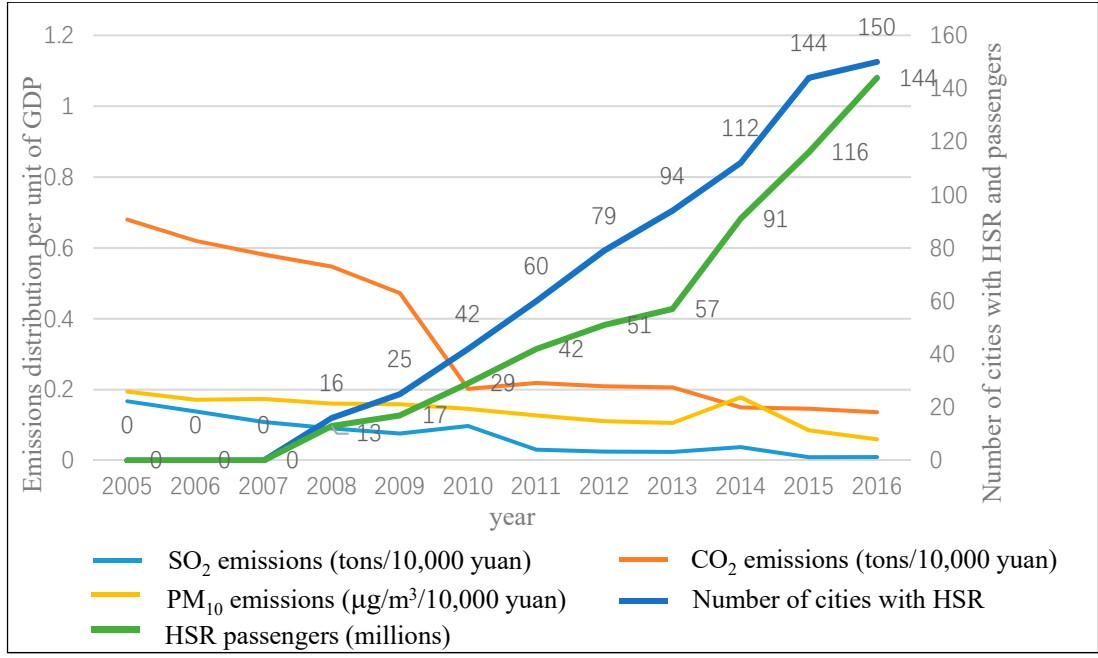

**Figure 2.** Pollution emissions and high-speed rail development.

Extending this line of research, this paper examines whether and how high-speed rail influences haze pollution based on institutional pressure. The most relevant literature with this paper is the study of Yang, et al. [13], who indicated that high-speed rail can improve production efficiency and resource utilization, as well as change industrial structure, all of which can help to reduce $SO_2$ emissions. This study investigated the effect of inter-city transportation on haze pollution. This is different from other pollutants and can be easily observed by ordinary individuals and, thus, has attracted more attention. In addition, we adopted an institutional pressure perspective to explain the effect of high-speed rail on haze pollution rather than an economic perspective for economic effect, which requires long-term testing. We propose that high-speed rail (HSR) operation can speed up crowd movement and information flow, which is beneficial for information transparency and can increase institutional pressure from investors, suppliers, media, superior supervision departments, tourists, and employers in the short term [14,15]. To reduce external pressure, governments, firms, and individuals have sought to reduce haze pollution through various behaviors that are able to gain legitimacy and social recognition [16].

To further verify the effectiveness of the underlying mechanisms, we examine two contingent factors that mitigate the effect of high-speed rail in the context of China and can strengthen or weaken its relationship with haze pollution. Such work can help identify the boundary conditions related to the impact of high-speed rail and can verify the mechanisms linking HSR to haze pollution. We hypothesize that the operation of high-speed rail increases the institutional pressure on haze pollution, due to information transparency. Thus, when the market development level is relatively undeveloped, which means that information asymmetry is serious, the operation of high-speed rail can have a stronger impact on the information environment. Hence, the operation of HSR can result in more institutional pressure on local cities and can lead to greater haze pollution reductions. Conversely, when the users' proportion of information communication technology (ICT) is already high, the operation of high-speed rail has a limited effect on information transparency and organizations experience less institutional pressure, which reduces the effect of HSR on haze pollution.

Our work contributes to three different branches of business research. First, we contribute insights into the mechanism for reducing haze pollution. Previous studies have mainly focused on urban transportation infrastructures [8,10]; this study supplements such work by investigating city–to–city transportation. Second, we contribute to transportation infrastructure research by recognizing the environmental and economic functions of high-speed rail. Previous studies have mainly focused on environmental pollution in the construction of transportation facilities [17] and the direct effects of various transportation modes, whereas this study emphasizes the indirect effect of environmental protection brought by the operation of traffic facilities. Third, we adopt the institutional pressure theory, thus bringing a fresh perspective to haze pollution research, which has primarily focused on planned behavior theory [18–21].

## 2. Theoretical Background

### 2.1. Haze Pollution Reduction Mechanisms

Haze pollution is a kind of synthetic contaminant that contains $SO_2$, NOx and other inhalable particles. Scholars typically use $PM_{10}$ and $PM_{2.5}$ to represent this kind of pollution [8]. Particulate matter below 10 microns in diameter is called $PM_{10}$, and can deposit in the upper respiratory system [22]. Similarly, particulate matter with a diameter of 2.5 microns or less is called $PM_{2.5}$, which can be deposited deeply into the lungs [19]. Haze pollution is different from other sources of pollution, such as $SO_2$, $CO_2$, and $NO_2$, because of its influence on visibility [1]. Haze pollution has attracted scholarly attention for decades, as it causes serious damage to individuals' health, as well as to local economies and reputations [4,5].

Therefore, a number of studies have focused on mechanisms to reduce haze pollution, including stricter regulations, technological innovations, and infrastructure improvements. Li, et al. [23]

documented that disclosing socially responsible behaviors in annual reports can put pressure on firms to reduce haze pollution under the regulation of government policies. For example, the Air Pollution Control Action has obvious deterrent and regulatory effects on haze pollution [19,24]. Ziyarati, et al. [25] highlighted that $CO_2$ and $NO_2$ emissions can be decreased based on processing plants of firms. Zhang and Wang [26] proved that energy-saving technologies can help to degrade pollutant emissions and improve product efficiency in the iron and steel sector. Zhao, et al. [3] argued that residents tend to purchase energy-saving appliances (instead of energy-consuming equipment) after haze pollution intensifies and that such changes can lead to a dramatic reduction of haze.

Infrastructure improvement is one of the most important reduction mechanisms, and previous studies have mainly focused on the effect of urban infrastructure on environmental pollution, whereas few studies have focused on the relationship between haze pollution and long-distance transportation. For instance, Luo, et al. [12] used road density to test transportation infrastructure and showed that a higher road intensity corresponded to a higher concentration of $PM_{10}$ during traffic jams. Chen and Whalley [10] found that pollution emissions were reduced after individuals reduced their use of motor vehicles. Li, et al. [11] also examined the effect of urban transportation and electric cars on environmental pollution. The mass implementation of electric cars can reduce the use of oil and other energies, thereby reducing the emission of pollutants.

High-speed rail, a new type of transportation infrastructure that connects various populations across provinces and cities, has provided an opportunity to study the effects of city–to–city transportation on haze pollution.

### 2.2. High-Speed Rail, Institutional Pressure, and Haze Pollution

High-speed rail is defined as trains that can achieve speeds exceeding 250 km/h with little delay; thus, they can shorten travel time, relative to other modes of transportation (except for air transportation). The first high-speed rail was introduced in China in 2008, and China currently has the longest high-speed rail track (in terms of mileage) in the world [27]. Lin [28] used a passenger survey and ridership data to find that HSR is quicker than conventional trains (Tekuai) and highway transportation in mid- and long-distance trips. In addition, HSR has very stable running times and is unlikely to cause bumps, which can increase the comfort of the journey [29].

The original purpose of building high-speed rail was to facilitate transportation; however, studies have revealed other key functions, such as gathering resources and increasing accessibility, and so on. For example, Willigers and van Wee [30] found that high-speed rail can help firms to select office locations, as HSR operations can promote the gathering of individuals and resources. Chen and Haynes [31] indicated that high-speed rail is beneficial to the development of the tourism industry, as it shortens travel times and increases comfort. Haynes [32] found that HSR aids labor market development. Levinson [33] also highlighted that HSR can increase the accessibility of customers, employees and suppliers, thus providing opportunities to local cities to promote economic development [34].

Scholars have also pointed out that high-speed rail can increase information communication and transparency. Hornung [35] indicated that it works as a convenient and fast means of transportation, which can promote the efficiency of information communication and can become the driving force for economic growth. Lin [28] also argued that 60% of riders take HSR for business purposes, and these individuals mainly belong to industries with nonroutine tasks, thus requiring employees with abstract skills, such as intuition, persuasion, and creativity, to conduct face-to-face communication. Tierney [36] held the opinion that HSR is an important infrastructure for the knowledge economy because it triggers knowledge and idea exchange.

In this study, we argue that the information transparency effect of high-speed rail also indirectly leads to institutional pressure on local governments and firms regarding haze pollution, due to its bad influence on talent outflow, economic development, and irresponsible image. To be more specific, environmental issues have become an important parameter among employees when considering their work address, as a poor environment leads to more health problems. Laumbach and Kipen [37]

suggested that air pollution leads to a large global burden of respiratory and allergic diseases, such as chronic obstructive pulmonary disease, asthma, and tuberculosis. Ahmed, et al. [38] highlighted that particulate matter can cause cardiovascular dysfunction, diabetes, and obesity. According to the analysis of the Global Burden of Disease Study [39], outdoor air pollution in China in 2010 caused 1.234 million premature deaths and 25 million cases of reduced health. In 2015, 163.1 deaths per 100,000 people were attributed to air pollution; thus, China has one of the highest air pollution-related disease burdens in the world. In addition, Zhang, et al. [2] assumed that haze pollution has an impact on tourist decision-making processes. If haze pollution in a destination city is serious, tourists may be more likely to change their travel plans to another, similar destination with good air quality. Nguitragool [4] also concluded that serious haze pollution can impair a country's reputation and lead to a loss of international credibility.

However, before the operation of high-speed rail, governments, firms, and individuals did not always engage in environmental protection activities, due to the distance. First, the distance between higher levels of government and local governments and enterprises leads to lax regulation and information asymmetry. According to the context of China, the higher levels of government are focused on environmental governance through policy-making, while local governments and firms may not comply with such policies when the higher levels of government are located far away from the local cities [40]. For enterprises, focusing on environmental protection leads to costs in terms of funds, resources, and manpower in the short term, resulting in a reduction of business operating profits [41]. Meanwhile, the decrease in corporate profits leads to a reduction in local government tax revenue [42]. Therefore, local governments and firms are reluctant to perform environmental-related activities when considering their own profits. Wang, et al. [43] showed that the various distances between listed Chinese firms and the central government have different influences on environmental protection strategies. With shorter distances, firms faced with stricter regulations will be inclined to adopt active environmental strategies.

Second, distance also causes information asymmetry and less pressure among stakeholders, local governments, and firms. Previous studies have shown that listed companies in less developed markets are more likely to conceal their true financial status. In those areas, investors, consumers, and suppliers have less channels to master the needed information and, hence, make less demands on the companies [44]. Lerner [45] found that investors preferred to choose closer locations as investment targets, in order to reduce supervision costs. The farther the investor is from the company, the higher the cost of collecting and processing information and the more serious the problem of information asymmetry. Hortaçsu, et al. [46] also posited that, in the process of online platform transactions, distance is also an important factor influencing the transaction choice between buyers and sellers, as sellers show less deceit over a relatively short distance. Kim, et al. [47] indicated that firms are willing to acquire companies that are closer to them. Over a shorter distance, firms have more opportunities to communicate with employees and consumers with targeted companies and perform field research to better evaluate and regulate the acquired companies.

Reducing haze pollution has become a common issue of concern, and the opening of high-speed rails has led to a better understanding among higher government levels and stakeholders about local cities and has allowed for increased demands in terms of pollution reduction. Therefore, according to institutional theory, information transparency leads to institutional pressure on local governments, firms, and individuals to focus on reducing haze. Therefore, we propose the following hypothesis:

**Hypothesis 1 (H1).** *The operation of high-speed rail has improved information transparency, which increases institutional pressure placed on organizations to reduce haze pollution.*

Although the operation of high-speed rail decreases information asymmetry and increases the institutional pressures placed on organizations to take measures to reduce haze pollution, the relationship between high-speed rail and haze pollution is expected to somewhat vary under

different conditions of the operational environment. We consider two potential factors that may influence this variance; namely, the development of hard infrastructure (information communication technology) and soft infrastructure (market development level), which reflect the information environment of the local area.

*Information communication technology:* A greater percentage of people now have access to mobile cellular technologies and the internet, which has inspired researchers to explore whether such technologies can improve economic and social development [48]. Sturges and Corruption [49] found that information communication technology promotes the information transparency of governments by removing the information access barrier and asymmetry. Gaskins [50] also found that the popularity of information communication technologies, such as the internet and mobile cellular devices, has an impact on decreasing government corruption. In addition, the study also extended that, when the popularity of information communication technology increased by 27%, information transparency could increase by 17.581%. Therefore, if the proportion of local users of information communication technology is larger, then the information environment is relatively transparent. In such cases, the operation of high-speed rails will have a lesser influence on information markets and haze pollution. Therefore, we propose the following hypothesis:

**Hypothesis 2 (H2).** *The negative relationship between high-speed rail and haze pollution can be weakened if the proportion of information communication technology users is higher.*

*Market development level:* In addition to hard infrastructure, the soft infrastructure market development level also has an influence on stakeholder awareness and organizational visibility. Generally, organizations located in relatively undeveloped markets are more enclosed and, thus, have lower visibility among the public and stakeholders [44]. The public cannot obtain much information about the haze pollution induced by these organizations from the media or other channels and, therefore, the organizations experience less pressure to deal with environmental problems. Thus, the operation of high-speed rail has a greater effect on the information environment and leads to more institutional pressure placed on organizations. In developed areas, the information environment is very mature and stakeholders have access to various information data about organizations in daily life and are accustomed to the local environmental situation; thus, the effect of HSR is reduced [50]. Therefore, we propose the following hypothesis:

**Hypothesis 3.** *The negative relationship between high-speed rail and haze pollution can be strengthened if the market development level is lower.*

## 3. Data and Sample

### 3.1. Data

We obtained prefecture-level socioeconomic data from city statistical yearbooks and regional economic statistical yearbooks in China. The operation time data for high-speed rails were collected and sorted by hand. Data missing in certain years were supplemented by the average growth rate method. We excluded observations where the main variables were missing and obtained unbalanced data for 3436 samples from 288 prefecture-level cities over 12 years (from 2005 to 2016), as many variables were missing before 2005.

### 3.2. Model and Variables

In the analysis process, our hypotheses were tested by generalized estimating equations (GEEs). This method is appropriate for the purposes of this paper, as it addresses unobserved heterogeneity between cities and the autocorrelation that results from the repeated measurement of these cities over time [51]. We specified a log link function, a Gaussian family, and an independent correlation structure for all GEE models, as the dependent variable is the logarithm and forms a Gaussian distribution.

In addition, we used robust standard errors to account for potential misspecifications of related structures and heteroscedasticity [52].

The operation time of high-speed rails for different cities varies from 2008 to 2016; thus, the conventional Difference-in-Difference (DID) method is not suited for this situation. Therefore, in this paper, we mainly adopted the time-varying DID to test the effect of high-speed rail. If the city opened an HSR during the period from 2008 to 2016, then du=1; otherwise, du=0. If the city opened an HSR in a certain year, then dt=1; otherwise, dt=0. According to the time-varying DID, the core variable DID = du ∗ dt, which represents the policy treatment effect. Therefore, the baseline regression model of this paper is as follows:

$$Pollution_{ct} = \beta_0 + \beta_1 DID_{ct} + \beta_2 Control_{ct} + \beta_3 C_c + year_t + \partial_i + \delta_{ct} \tag{1}$$

where *Pollution$_{ct}$* represents the environmental pollution indicators, with the concentration of $PM_{10}$ adopted as the proxy variable ; *DID* represents du ∗ dt; *Control* is a set of control variables; *year* represents the year fixed effect; *α* represents the city-specific fixed effect; *δ* is the error term; and $\beta_0, \beta_1, \beta_2$, and $\beta_3$ are the coefficients to be estimated. The control variables and measurement method are defined as follows, and Table 1 shows all the variables and their definitions.

Market development level (Govexpend). The government expenditure as a percentage of gross domestic product (GDP) is high, which means that the level of market development is relatively low and the information environment is opaque [44]. This paper used government expenditures divided by GDP to measure the market development level.

Information communication technology (Phonepeople). The popularity of information communication technology represents the information transparency degree [48]. This paper used the number of mobile users divided by the total population to measure the popularity of information communication technology.

Government regulation (Govregulate). Many studies have proved that government regulation has an influence on firm environmental pollution [40,53,54]. This paper used the proportion of investment in industrial pollution control divided by GDP to measure the intensity of government regulation.

Oil supply (Lnoilhome). The use of oil can release pollutants and can influence the formation of environmental pollution [55]. This paper used the total oil supply divided by the total population to measure the oil demands.

Gas supply (Lngashome). Burning natural gas can also generate some pollutants, such as $CO_2$ and NOx, which has an impact on haze pollution [56]. This paper used the total gas supply divided by the total population to measure the gas demands.

Public transportation (Lnpublictrans). The use of public transportation can reduce the use of private cars as well as pollution emissions [57]. This paper adopted the number of public buses per 10,000 people to represent the public transportation situation.

Industrial structure (secondgdp, secondemploy). Secondary industries are the main source of environmental pollution, due to the high number of manufacturing firms in this industry. This paper used secondary industry GDP divided by total GDP and secondary industry employees in the total population to reflect the impact of the industry structure on haze pollution.

Openness level (Lnfdi). Various studies have examined the relationship between foreign direct investment (FDI) and environmental pollution and have indicated that the FDI can increase environmental pollution, based on the pollution heaven theory [58]. Therefore, this paper used the total FDI investments divided by GDP to represent the openness level.

Economic level (Lnpergdp, Lnaveragepay). Numerous studies have shown that there are many relationships between environmental pollution and economic level, such as the inverted U–shaped trend and the N–shaped trend [59]. This paper used the per capital GDP and average wage of workers to represent the economic level.

Education conditions (Lnnumhistu). Education conditions have a considerable influence on environmental pollution. Kan, et al. [60] studied the relationship between education and air pollution,

and the results showed that air pollution is more serious in areas with lower education level areas. This paper used the proportion of high-school students in the total population to represent the education conditions.

Technological development situation (Lnsciemplo). Technological development is an important indicator for reflecting environmental pollution, with a better technological development corresponding to greater resource utilization efficiency and lower pollution discharge [13]. This paper used the proportion of employees in the scientific and technological industries among the total population to represent the technological development level.

Population density (Lnpopdensity). Due to the large differences in population size and administrative area among cities, the population numbers in each city have no comparable value. This paper used the population density (i.e., the number of people per unit area in an administrative region) to represent the impact of population on environmental pollution [13].

**Table 1.** Definition of variables.

| Variables | Definitions |
| --- | --- |
| $LnPM_{10}$ | Haze score ($\mu g\ /m^3$) |
| WheHSR | HSR dummy: If the city opens an HSR in the observation year, HSR = 1; otherwise, HSR = 0. |
| Govexpend | Government expenditures/GDP (%) |
| Phonepeople | Number of mobile users/total population (%) |
| Govregulation | Investment of industrial pollution control/GDP (%) |
| Lnoilhome | Total oil supply/total population (tons/10,000 people) |
| Lngashome | Total gas supply/total population (tons/10,000 people) |
| Lnpublictrans | Public buses per 10,000people (unit) |
| Lnaveragepay | Average wage of workers (yuan) |
| Secondgdp | Output value of the secondary industry/GDP (%) |
| Lnfdi | Actual foreign investment/GDP (%) |
| Lnpergdp | Per capita GDP (yuan) |
| Lnnumhistu | Number of high school students/total population (%) |
| Lnsciemplo | Number of scientific research employees/total population (%) |
| Secondemploy | Number of secondary industry employees/total population (%) |
| Lnpopdensity | Population density (10,000people/$km^2$) |

Note: Ln stands for logarithm.

## 4. Results

### 4.1. Descriptive Analysis

Table 2 shows the descriptive analysis of the treatment group (HSR = 1) and control group (HSR = 0) in our final sample. The results show that the mean value of $PM_{10}$ in the treatment group (12.42) was less than that in the control group (12.92) and ranged from 2.77 to 16.92 $\mu g\ /m^3$, with a standard deviation of 4.13. In addition to the proportion of secondary industry GDP and other variables, the mean values of the treatment group were higher than that of the control group. Compared with previous studies, the distribution of variables did not make much a difference [13], which ensures the quality and precision of the data.

**Table 2.** Descriptive statistical analysis.

| | | HSR=1 | | | | HSR=0 | | | |
|---|---|---|---|---|---|---|---|---|---|
| Variable | Observation | Mean | St.Er | Min | Max | Mean | St.Er | Min | Max |
| $LnPM_{10}$ | 3409 | 12.42 | 4.13 | 2.77 | 16.92 | 12.92 | 2.60 | 2.77 | 16.92 |
| Govexpend | 3395 | 0.30 | 0.22 | 0.08 | 2.43 | 0.28 | 0.30 | 0.07 | 4.94 |
| Phonepeople | 3433 | 56.73 | 705.11 | 0.03 | 11841.22 | 46.10 | 260.98 | 0.02 | 7590.31 |
| Lnoilhome | 3106 | 3.72 | 1.41 | 0.01 | 7.23 | 2.74 | 1.52 | 0.01 | 8.33 |
| Lngashome | 3326 | 3.62 | 1.38 | 0.01 | 7.96 | 3.30 | 1.33 | 0.15 | 8.58 |
| Lnpublictrans | 2862 | 2.26 | 0.59 | 0.77 | 5.42 | 1.90 | 0.60 | 0.28 | 4.65 |
| Lnaveragepay | 3434 | 10.85 | 0.32 | 9.56 | 11.70 | 10.38 | 0.44 | 8.51 | 11.70 |
| Secondgdp | 3154 | 48.43 | 10.18 | 18.57 | 79.36 | 49.41 | 10.92 | 15.70 | 89.34 |
| Lnfdi | 3282 | 0.00 | 0.00 | 0.00 | 0.03 | 0.01 | 0.01 | 0.01 | 0.17 |
| Govregulate | 3407 | 2.40 | 0.56 | 0.68 | 4.34 | 2.69 | 0.64 | 0.68 | 4.61 |
| Lnpergdp | 3154 | 10.76 | 0.57 | 9.09 | 12.28 | 10.17 | 0.70 | 7.66 | 12.28 |
| Lnnumhistu | 3321 | 11.24 | 1.14 | 8.02 | 13.78 | 10.17 | 1.30 | 5.45 | 13.78 |
| Lnsciemplo | 3434 | 2.05 | 1.64 | 0.01 | 5.50 | 0.63 | 1.18 | 0.01 | 5.11 |
| Secondemploy | 3436 | 48.28 | 13.82 | 8.12 | 83.30 | 43.26 | 14.17 | 1.77 | 83.30 |
| Lnpopdensity | 2865 | 6.15 | 0.67 | 3.83 | 7.88 | 5.65 | 0.93 | 1.57 | 7.87 |

Table 3 shows the pairwise correlations among the variables in our sample. Almost all variables were correlated with haze pollution, which proves that the selected variables are rational. To test whether there was a collinearity problem among variables, a collinearity test was performed. The mean variance inflation factor (VIF) score for the variables was 2.02, the highest score was 5.67, and no scores were higher than 10. Therefore, there is no collinearity concern in this study.

**Table 3.** Correlation matrix.

| | LnPM10 | 1 | 2 | 3 | 4 | 5 | 6 | 7 | 8 | 9 | 10 | 11 | 12 | 13 | 14 |
|---|---|---|---|---|---|---|---|---|---|---|---|---|---|---|---|
| 1.WheHSR | −0.07 | 1.00 | | | | | | | | | | | | | |
| 2.Govexpend | −0.48 | 0.03 | 1.00 | | | | | | | | | | | | |
| 3.Phonepeople | 0.01 | −0.07 | −0.07 | 1.00 | | | | | | | | | | | |
| 4.govregulate | 0.06 | −0.18 | −0.07 | 0.11 | 1.00 | | | | | | | | | | |
| 5.Lnoilhome | 0.06 | 0.27 | −0.12 | 0.00 | −0.04 | 1.00 | | | | | | | | | |
| 6.Lngashome | 0.05 | 0.10 | −0.21 | 0.13 | −0.22 | 0.28 | 1.00 | | | | | | | | |
| 7.Lnpublictrans | 0.36 | 0.21 | −0.24 | 0.04 | −0.11 | 0.47 | 0.37 | 1.00 | | | | | | | |
| 8.Lnaveragepay | −0.07 | 0.42 | 0.27 | −0.19 | −0.34 | 0.41 | 0.14 | 0.35 | 1.00 | | | | | | |
| 9.Secondgdp | 0.27 | −0.04 | −0.35 | 0.03 | 0.04 | 0.23 | 0.06 | 0.20 | 0.06 | 1.00 | | | | | |
| 10.Lnfdi | −0.07 | 0.00 | 0.05 | 0.00 | −0.05 | 0.00 | 0.06 | 0.01 | −0.03 | 0.00 | 1.00 | | | | |
| 11.Lnpergdp | 0.49 | 0.32 | −0.22 | −0.06 | −0.27 | 0.60 | 0.43 | 0.59 | 0.73 | 0.37 | 0.01 | 1.00 | | | |
| 12.Lnnumhistu | 0.17 | 0.33 | −0.14 | 0.00 | −0.14 | 0.42 | 0.29 | 0.51 | 0.28 | −0.05 | 0.01 | 0.42 | 1.00 | | |
| 13.Lnsciemplo | −0.17 | 0.41 | 0.21 | −0.11 | −0.10 | 0.29 | 0.03 | 0.18 | 0.68 | −0.14 | −0.03 | 0.43 | 0.19 | 1.00 | |
| 14.Secondemploy | 0.23 | 0.14 | −0.29 | 0.03 | −0.09 | 0.37 | 0.25 | 0.36 | 0.19 | 0.57 | −0.03 | 0.51 | 0.21 | 0.07 | 1.00 |
| 15.Lnpopdensity | 0.16 | 0.20 | −0.27 | 0.04 | −0.27 | 0.16 | 0.23 | 0.15 | 0.09 | 0.12 | 0.01 | 0.16 | 0.47 | 0.02 | 0.39 |

### 4.2. Baseline Regression of the Time–Varying DID Method

The presupposition of the DID model is that the development trend between the treatment group and control group is parallel. Therefore, we conducted a parallel trend test between the control group and treatment groups. According to the method of Beck, et al. [61], we used the interaction of the dummy variable and the operation of high-speed rail at each point in time. If the coefficient of the interaction term before the operation of high-speed rail is not significant, then this represents that there is a parallel trend between the control group and treatment group. In this study, we considered an 18–year window, spanning from 8 years before the operation of HSR to 10 years after the operation of HSR. Figure 3 describes the results, and the dashed lines represent 95% confidence intervals (adjusted for city-level clustering). The estimated coefficients were obtained from the Equation (2). In the Equation (2), $H_{ct}^{-j}$ equals 1 for cities in the j years before the operation of HSR and $H_{ct}^{+j}$ equals 1 for cities in the *j* years after the operation of HSR. We excluded the year of HSR operations and estimated

the dynamic effects of operation on haze pollution. $A_c$ and $B_t$ represent the city and year dummy variables, which account for the city and year fixed effects, respectively.

$$Pollution = \alpha + \beta_1 H_{ct}^{-8} + \beta_2 H_{ct}^{-7} + \cdots + \beta_{18} H_{ct}^{+10} + A_c + B_t + \varepsilon_{st} \tag{2}$$

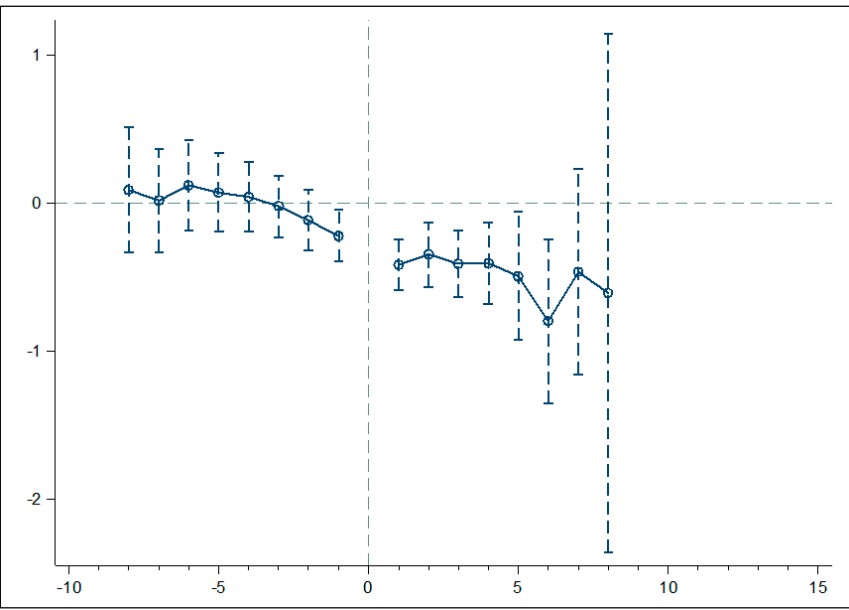

**Figure 3.** Parallel trend test coefficient diagram.

Figure 3 predicts two important issues: The changes in haze pollution did not precede the operation of HSR, and the impact of HSR operations on the haze pollution materialized rapidly. As shown, haze pollution increased immediately after HSR operations, such that H+1 was negative and significant at the 5% level. Thus, the particular mechanisms and channels that connected HSR operations with haze pollution must be fast acting. Therefore, changes in haze pollution do not precede HSR operations and HSR operations have a level effect on haze pollution.

After the parallel tend test, Table 4 reports the baseline results for our three hypotheses regarding the contingent impacts. Model 1 in Table 4 only includes the control variables. Model 2 tests the main effect of HSR operations on haze pollutionand adds city and year dummy variables while controlling for all other independent variables. Models 3 and 4 test the moderating effects of information communication technology, while Models 5 and 6 also examine the moderating effects of the market development level, and the difference is whether the interaction term is controlled.

As can be seen from Model 2, HSR had a significant influence on haze pollution, as after the operation of high-speed rail, the concentration of haze pollution decreased by 17%. In Model 4, the popularity of information communication technology had a positive influence on the negative relationship between high-speed rail and haze pollution, which means that a greater number of users of information communication technology corresponds to a more transparent information market, which decreases the negative effect of high-speed rail. In Model 6, the market development level had a negative influence on the relationship between high-speed rail and haze pollution, which means that the market is more undeveloped and the information environment is more asymmetric; thus, the effect of high-speed rail is more significantly negative on haze pollution.

**Table 4.** Baseline regression results of the time-varying difference-in-difference (DID) model.

| Variables | Model1 | Model2 | Model 3 | Model 4 | Model 5 | Model 6 |
|---|---|---|---|---|---|---|
| DID | | −0.17 *** | −0.19 *** | −0.16 ** | −0.18 *** | −0.20 *** |
| | | (−2.78) | (−2.76) | (−2.23) | (−2.72) | (−2.60) |
| DID*Lnphonepeople | | | | 0.07* | | |
| | | | | (1.78) | | |
| Lnphonepeople | | | 0.17 ** | 0.16 ** | | |
| | | | (2.44) | (2.32) | | |
| DID*Govexpend | | | | | | −0.21 * |
| | | | | | | (−1.68) |
| Govexpend | | | | | 0.16 | 0.18 |
| | | | | | (0.90) | (0.94) |
| Govregulate | 0.07 | 0.08 | 0.08 | 0.08 | 0.07 | 0.07 |
| | (1.28) | (1.44) | (1.50) | (1.50) | (1.33) | (1.33) |
| Lnoilhome | 0.03 | 0.03 | 0.02 | 0.01 | 0.03 | 0.03 |
| | (0.85) | (0.95) | (0.50) | (0.47) | (1.05) | (1.05) |
| Lngashome | −0.04 | −0.04 | −0.06 * | −0.06 ** | −0.05 | −0.05 |
| | (−1.33) | (−1.40) | (−1.93) | (−1.97) | (−1.48) | (−1.46) |
| Lnpublictrans | 0.08 | 0.09 | 0.08 | 0.08 | 0.09 | 0.09 |
| | (1.18) | (1.33) | (1.20) | (1.17) | (1.29) | (1.29) |
| Lnaveragepay | 0.40 * | 0.43 ** | 0.41 * | 0.41 * | 0.38 * | 0.38 * |
| | (1.90) | (2.03) | (1.92) | (1.95) | (1.78) | (1.79) |
| Secondgdp | 0.02 *** | 0.02 *** | 0.02 *** | 0.02 *** | 0.02 *** | 0.02 *** |
| | (4.38) | (4.32) | (4.41) | (4.44) | (4.20) | (4.21) |
| Lnfdi | 1.18 | 1.23 | 1.25 | 1.21 | 1.09 | 1.07 |
| | (0.65) | (0.67) | (0.68) | (0.66) | (0.60) | (0.58) |
| Lnpergdp | 0.24 * | 0.24 * | 0.18 | 0.17 | 0.26 ** | 0.26 ** |
| | (1.87) | (1.86) | (1.38) | (1.32) | (2.02) | (2.02) |
| Lnnumhistu | 0.24 *** | 0.23 *** | 0.23 *** | 0.24 *** | 0.23 *** | 0.24 *** |
| | (4.48) | (4.57) | (4.57) | (4.58) | (4.55) | (4.56) |
| Lnsciemplo | −0.14 *** | −0.13 *** | −0.14 *** | −0.14 *** | −0.13 *** | −0.13 *** |
| | (−2.92) | (−2.74) | (−2.77) | (−2.91) | (−2.76) | (−2.76) |
| Secondemploy | 0.01 | 0.01 | 0.01 | 0.01 | 0.01 | 0.01 |
| | (0.50) | (0.51) | (0.58) | (0.65) | (0.53) | (0.52) |
| Lnpopdensity | −0.07 | −0.06 | −0.05 | −0.06 | −0.06 | −0.06 |
| | (−0.96) | (−0.81) | (−0.78) | (−0.82) | (−0.88) | (−0.88) |
| Constant | 3.11 | 2.73 | 2.23 | 2.30 | 3.01 | 2.97 |
| | (1.58) | (1.39) | (1.13) | (1.16) | (1.52) | (1.50) |
| Year fixed | Yes | Yes | Yes | Yes | Yes | Yes |
| N | 2327 | 2327 | 2327 | 2327 | 2313 | 2313 |
| Wald Chi2 | 1018.87 | 1030.55 | 1038.77 | 1043.50 | 1023.89 | 1023.99 |

Note: z−statistics in parentheses *** p < 0.01, ** p < 0.05, * p < 0.1.

## 4.3. Endogenous Treatment

In this paper, we regarded the opening of high-speed rail as an exogenous shock, while some studies also proposed some factors that may influence the HSR opening, such as city size, economic level, and human capital [13,34]. Therefore, the opening of high-speed rail may be an endogenous variable. To reduce this bias, we used the instrument variable method to test for endogenous problems. The instrument variable method is a useful and common method for identifying the impact of endogenous problems. In previous studies, climate characteristics, geographic slope, and historical information were used as instrument variables for transportation infrastructure [13,62].

We chose two variables to construct the instrument variable for HSR: The first one is the time difference of the operation of conventional train tracks and high-speed rail (trainyear), and the second one is the latitude of the local city. Both variables met the two conditions of correlation and exogenous. Therefore, our choice of instrument variable was rational.

In order to correct some econometric problems traditionally related with diversification effects such as sample selection bias, we applied the Heckman's two-stage method [63]. In the first stage, the procedure estimated the selection equation as a Probit model to analyze the propensity to open high-speed rail and calculated the Inverse Mills Ratio (lambda). The regression model was as follows. Where *wheHSR* represents whether the city had opened a high-speed railway in the fiscal year. *Control* is a set of all variables, *trainopyear* is the instrument variable, which was measured by the difference of the operation of conventional tracks and high-speed rail. $\delta$ is the normal error term.

$$wheHSR_{ct} = \beta_0 + \beta_1 control_{ct} + \beta_2 trainopyear + \delta_i \tag{3}$$

In the second stage, the corrected regression equation was estimated by GEE regression to examine the effects of high-speed railway on haze pollution. After incorporating the Inverse Mills Ratio, the final regression was as follows:

$$Pollution_{ct} = \begin{aligned}&\beta_0 + \beta_1 wheHSR_{ct} + \beta_2 lambda + \beta_3 Control_{ct} + \beta_4 C_c \\ &+ year_t + \partial_i + \delta_{ct}\end{aligned} \tag{4}$$

where *Pollution* represented the environmental pollution indicators, with the concentration of $PM_{10}$ adopted as the proxy variable; *wheBA* represents whether the city had opened a high-speed railway in the fiscal year; *lambda* is the Inverse Mills Ratio; *Control* is a set of control variables; *year* represents the year fixed effect; $\alpha$ represents the city-specific fixed effect; $\delta$ is the error term; and $\beta_0$, $\beta_1$, $\beta_2$, and $\beta_3$ are the coefficients that were to be estimated.

Table 5 reports the results of the instrumental variables. Model 1 is the first stage, when trainyear was adopted as the instrument variable. Model 2 is the second stage. Model 3 is the first stage, when latitude of the local city was adopted as the instrument variable, Model 4 is the second stage. The results were consistent with H1: the opening of high-speed rail has a negative influence on haze pollution.

**Table 5.** Endogenous test: based on Heckman's two-stage method.

| | Model1 | Model2 | Model3 | Model4 |
|---|---|---|---|---|
| **VARIABLES** | **WheHSR** | **LnPM10** | **WheHSR** | **LnPM10** |
| WheHSR | | −0.23 *** | | −0.16 ** |
| | | (−3.13) | | (−2.35) |
| Lambda | | 2.08 *** | | −1.11 ** |
| | | (4.50) | | (−2.02) |
| Trainyear | −0.01 *** | | | |
| | (−3.93) | | | |
| Latitude | | | 0.01 | |
| | | | (0.34) | |
| Govregulate | 0.08 | 0.19 *** | −0.01 | 0.08 |
| | (0.95) | (2.82) | (−0.17) | (1.60) |
| Lnoilhome | 0.18 *** | 0.31 *** | 0.09 *** | −0.06 |
| | (5.07) | (4.21) | (2.97) | (−1.13) |
| Lngashome | −0.05 | −0.12 *** | 0.03 | −0.07 ** |
| | (−1.47) | (−2.96) | (0.81) | (−2.15) |
| Lnpublictrans | 0.12 | 0.24 *** | 0.11 | −0.02 |
| | (1.42) | (2.63) | (1.43) | (−0.24) |
| Lnaveragepay | 0.46 * | 1.30 *** | 0.16 | 0.22 |
| | (1.74) | (4.29) | (0.68) | (0.99) |
| Secondgdp | −0.01 | 0.02 *** | 0.01 | 0.01 *** |
| | (−0.35) | (4.20) | (0.49) | (3.36) |

**Table 5.** *Cont.*

|  | Model1 | Model2 | Model3 | Model4 |
|---|---|---|---|---|
| **VARIABLES** | WheHSR | LnPM10 | WheHSR | LnPM10 |
| Lnfdi | −1.40 | 0.64 | 2.43 | −2.27 |
|  | (−0.29) | (0.23) | (0.60) | (−1.05) |
| Lnpergdp | −0.09 | −0.23 | −0.10 | 0.37 *** |
|  | (−0.74) | (−1.36) | (−0.86) | (2.72) |
| Lnnumhistu | 0.02 | 0.29 *** | 0.18 *** | 0.07 |
|  | (0.43) | (4.15) | (4.66) | (0.71) |
| Lnsciemplo | −0.08 | −0.33 *** | −0.03 | −0.06 |
|  | (−1.00) | (−4.84) | (−0.34) | (−1.10) |
| Secondemploy | −0.01 ** | −0.01* | −0.01 | 0.01 |
|  | (−2.39) | (−1.82) | (−1.48) | (1.58) |
| Lnpopdensity | 0.19 *** | 0.29 ** | 0.27 *** | −0.31 ** |
|  | (2.71) | (2.51) | (4.63) | (−2.22) |
| Constant | −6.49 *** | −9.32 *** | −6.65 *** | 9.85 ** |
|  | (−2.98) | (−2.71) | (−3.25) | (2.52) |
| Year Fixed | Yes | Yes | Yes | Yes |
| N | 1563 | 1537 | 2341 | 2307 |
| Wald Chi2 | 453.73 | 700.56 | 516.49 | 1048.7 |

Note: z−statistics in parentheses *** $p < 0.01$, ** $p < 0.05$, * $p < 0.1$

## 4.4. Robustness Checks

In order to test the robustness of results, we conducted four robustness checks to verify the robustness of the results. First, we considered other emissions. Haze pollution is a serious pollutant, and the operation of HSR has demonstrated a negative effect on it. However, what about other kinds of pollutants? We chose $CO_2$ and $SO_2$ as substitute variables to check the HSR effect, using Models 1 and 2 of Table 6, respectively. We can see that high-speed railway also had a negative effect on $CO_2$ and $SO_2$ emissions.

Second, we considered the large differences in the administrative levels of cities. Most cities in the sample are ordinary prefecture-level cities, although some municipalities and autonomous regions, such as Chongqing, Beijing, Tianjin and Shanghai, were also included. Other factors may influence environmental pollution in these special zones, due to their economic policies and administrative levels. Therefore, we deleted these special cities. Models 3 and 4 of Table 6 show the results without municipalities and without municipalities and autonomous regions, respectively. The coefficient of the HSR effect was also significantly negative, which supports the main hypothesis of this paper.

Third, in order to examine whether the HSR effect on haze pollution changes significantly with different sample time spans, we changed the time bandwidth. In the benchmark of this paper, the time bandwidth was from 2005 to 2016, and, therefore, we set the alternate time bandwidths to be from 2007 to 2011 and 2005 to 2013; the regression results using these bandwidths are shown in Models 5 and 6 of Table 6. The coefficients of HSR were also negative and significant, which indicates that the results of this paper are robust.

Finally, in order to verify the mechanism, we used another two moderating variables to represent the market development level and information communication technology. We used the proportion of private employees among the total population to represent the market development level: the higher the proportion, the more developed the market. In addition, information communication technology was measured by the proportion of internet users among the total population. Models 7 and 8 of Table 6 report the moderating effects of the popularity of information communication technology and the market development situation, respectively. The results were consistent with Hypotheses 2 and 3, which prove that the moderating effect is valid.

**Table 6.** Robust regression.

| Variables | Model1 | Model2 | Model3 | Model4 | Model5 | Model6 | Model7 | Model8 |
|---|---|---|---|---|---|---|---|---|
| | $CO_2$ | $SO_2$ | without Municipalities | Without Municipalities & Autonomous | 2007–2011 | 2005–2013 | $LnPM_{10}$ | $LnPM_{10}$ |
| DID | −0.05 ** | −0.03 | −0.17 *** | −0.13 * | −0.22 ** | −0.22 *** | −0.11 | −0.23 *** |
| | (−2.12) | (−0.66) | (−2.42) | (−1.82) | (−2.35) | (−2.81) | (−1.24) | (−2.78) |
| DID*Lninternetpeople | | | | | | | 0.04 ** | |
| | | | | | | | (2.50) | |
| Lninternetpeople | | | | | | | 0.05 * | |
| | | | | | | | (1.65) | |
| DID*Lnemployee | | | | | | | | 0.01 |
| | | | | | | | | (0.10) |
| Lnemployee | | | | | | | | −0.05 |
| | | | | | | | | (−1.01) |
| Constant | −2.40 *** | 7.76 *** | 3.47 * | 3.59 * | −0.67 | 1.41 | 1.34 | 1.43 |
| | (−2.66) | (6.30) | (1.73) | (1.77) | (−0.25) | (0.67) | (0.63) | (0.67) |
| Control | Yes | Yes | Yes | Yes | Yes | Yes | Yes | Yes |
| Year fixed | Yes | Yes | Yes | Yes | Yes | Yes | Yes | Yes |
| N | 1761 | 2345 | 2311 | 2119 | 1439 | 2080 | 2077 | 2071 |
| Wald Chi2 | 850.96 | 843.98 | 1008.74 | 955.51 | 565.16 | 940.64 | 1047.21 | 1028.4 |

Note: z−statistics in parentheses *** $p < 0.01$, ** $p < 0.05$, * $p < 0.1$.

## 5. Conclusions and Discussion

We use data from 288 prefecture-level cities between 2005 and 2016, the period associated with the operation of high-speed rail. We found that the operation of HSR significantly reduced the concentration of haze pollution, and that effect was stronger in areas with a lower popularity of ICT and lower level of market development and weaker in areas with higher popularity of ICT and higher market development level. These results show that the mechanism proposed by this study is correct: the operation of high-speed rail has increased the flow of individuals and information, as well as transparency. Information transparency leads to institutional pressure placed on local governments and firms regarding haze pollution. To reduce this institutional pressure, local governments and firms must redesign their strategic planning to conform to the demands of stakeholders to reduce the haze pollution.

Our results provide certain theoretical contributions. First, we supplement haze pollution research. Previous studies have focused on the formation of haze pollution, the construction of haze pollution, and the reduction mechanisms for haze pollution. Although, previously identified reduction mechanisms mainly include individual behaviors, firm measurements, government policies, and urban transportation infrastructures. We studied the relationships between city–to–city transportation infrastructure and haze pollution from an institutional perspective, thereby filling a research gap about haze pollution. Second, previous studies have mainly concentrated on the direct effects of transportation infrastructure on environment pollution, whereas few studies have focused on the indirect effects. For example, various studies have found that urban transportation can reduce environmental pollution by saving energy and reducing traffic jams. In contrast, this study pays attention to the indirect effects of transportation infrastructure on environmental pollution.

Our results also generate important managerial and policy implications. First, attracting the focus of stakeholders can provide an effective means to force organizations to focus on environmental problems. In recent years, a central issue in the Chinese economic development process is how to maintain sustainable development. However, policies issued by the higher levels of government cannot resolve this issue. For example, manufacturing industries often generate high profits based on high-pollution processes, and local governments often turn a blind eye to those firms to promote the economic development of the local region. In contrast, organizations have focused tremendous attention and efforts to satisfy the demands of stakeholders to gain competitive advantages. Therefore, taking measures to attract the focus of stakeholders is an effective way to regulate organizations to protect the environment.

Second, the market development level is beneficial to the development of local regions by improving the development of the economy and by attracting high-quality employees and new technologies, and the effect on environmental pollution is no exception. At a higher market development level, organizations seeking to attract more attention and market share must meet the expectations of stakeholders. In contrast, in undeveloped areas, the market environment is moderate, there are less competitors to consider, and there is less pressure to care about the demands of stakeholders. Therefore, improving the market development environment is an important task for governments and prioritizing the market is the most effective means of achieving sustainable development.

The potential limitations of this study are threefold: First, we only focused on the effects of the operational phrase of high-speed rail on pollution emission, not considering the other stages of the railway field. For example, the recycling and recovery of metro trains has an influence on environmental protection [64,65]. Second, due to the availability of transportation infrastructure, we do not consider robustness tests for other modes of transportation, such as highways and aircraft. Future research will focus on the relationships between other transportation infrastructures and haze pollution. Third, in this paper, we only consider the moderating effect instead of the mediating effects, such as the measurement of institutional pressure. Future studies will combine the moderating and mediating effects to study the overall effect of transportation infrastructure on haze pollution.

**Author Contributions:** The study was a collaboration of three people. Conceptualization, Y.W.; methodology, Y.C.; Writing, Y.C. and R.H. All authors have read and agreed to the published version of the manuscript.

**Funding:** This research was founded by the National Natural Science Foundation of China (# 71974136).

**Conflicts of Interest:** The authors declare no conflict of interest.

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
