# Peer review of "Sustainability by High–Speed Rail: The Reduction Mechanisms of Transportation Infrastructure on Haze Pollution"

_sustainability, doi:10.3390/su12072763_

Round 1
Reviewer 1 Report
First of all, the authors should improve the quality of text, especially the English!
The title sound quite bad, you should use "Sustainability by High..." instead of "with". Professional English correction is required!
The paper is quite similar to "Can high-speed rail reduce environmental pollution? Evidence from China", by Xuehui Yang et al 2019. The authors should clearly underline the novelty of their paper (Abstract) and the main differences considering similar papers.
The authors should use the correct form of SO2, NO2, etc.
I suggest the authors to add more information regarding the quantity of emissions. They could do a plot to show the emissions evolution, number of HSR passengers, number of car in China for 2005-2016.
Too many tables, try to convert some of them in Figures.
THE TEXT SHOULD BE MORE EXPLICIT, CONCISE AND SHORT!
Where is Figure 1? I suggest for Figure 1 a map of China with HSR network and some info about pollution? (if possible, you could use data (a map) from OMI or TROPOMI www.temis.nl or nasa giovanni)
If possible add more Figures, to see trends or evolutions of some indexes mentioned in your paper.
Burning gas you can not get SO2!, please correct! (Line 281)
Author Response
Thanks for giving us the unique opportunity to revise our work. It means a lot to us.
In short, we mainly revised this piece of work in two ways. First, we strengthened our empirical work with an effort to handle the endogeneity issues (i.e., selection bias). Second, we improved the language by using the MDPI copyediting service.
Point1: First of all, the authors should improve the quality of text, especially the English.
Response1: We have adopted the MDPI copyediting service to improve the language.
Point2: The title sound quite bad, you should use "Sustainability by High..." instead of "with". Professional English correction is required.
Response2: Thanks, we have modified as you suggested.
Point3: The paper is quite similar to "Can high-speed rail reduce environmental pollution? Evidence from China", by Xuehui Yang et al 2019. The authors should clearly underline the novelty of their paper (Abstract) and the main differences considering similar papers.
Response3: Yes, Yang et al.’s work opened a promising avenue to investigate the effect of HSR on local economy, environment and other issues. In this new avenue, we have the similar focus on the effect on environment pollution. While, there are still at least two aspects that differ our work from theirs. First, we focus on the haze pollution as the main dependent variable. We believe this is more visible for people, particular for individuals that might take HSR, and turn out exerts effect of local authority, firms and other entities. Thus, it is much direct and observable to capture the HSR effect. Second, the underlying mechanism used in the present study differs. To be consistent with the flows of people and information, we believe the institutional pressure is much relevant to base studies investigating the HSR effect, particularly in the short term. To highlight these two points, we put them into the abstract and introduction.
Point 4: The authors should use the correct form of SO2, NO2, etc.
Response4: We corrected them as SO2, NO2, PM10, PM2.5.
Point5: I suggest the authors to add more information regarding the quantity of emissions. They could do a plot to show the emissions evolution, number of HSR passengers, number of car in China for 2005-2016.
Response5: Thanks for your suggestion. We believe these are very useful to demonstrate the latest development of HSR in China, and the intuitive impression of the relationship between the HSR development and pollution. Follow your suggestion, we added a map of China’s latest HSR network, and a figure to show the development of HSR in terms of HSR passengers and cities that have operated HSRs, and their relationships with pollution proxied by SO2, CO2, PM10.
Point6: Too many tables, try to convert some of them in Figures.
Response6: Thanks for your good suggest. In the current version, we have merged table7, table 8 into table 6. In all, we all have 6 tables and 3 figures.
Point7: THE TEXT SHOULD BE MORE EXPLICIT, CONCISE AND SHORT.
Response7: Thanks, we are sorry for this. In the current version, we have tried to make our sentences more concise and short. We tried to figure out one key messages for each paragraph, and cut down some reduction and repeat expressions. The copyeditor also helped us to perfect our language. We hope this is in line with your expectation.
Point8: Where is Figure 1? I suggest for Figure 1 a map of China with HSR network and some info about pollution? (if possible, you could use data (a map) from OMI or TROPOMI www.temis.nl or nasa giovanni).
Response8: Thanks, we have added them as you suggested.
Point9: If possible add more Figures, to see trends or evolutions of some indexes mentioned in your paper.
Response9: We have updated the indexes of tables and figures.
Point10: Burning gas you cannot get SO2, please correct. (Line 281).
Response10: Sorry for that mistake, we have revised it.Line827.
Reviewer 2 Report
This paper studies the effect of high speed train on haze pollution and provides evidence that it reduces haze pollution.
concerns of this paper are:
The authors suggest for Hypothesis 1, that the "operation of high-speed rail has improved information transparency, which increases
institutional pressure on organizations to reduce haze pollution." It is not clear what the authors' argument is in terms of making causal inference based on a model.
The authors argue that although the operation of high-speed rail decreases information asymmetry and increases institutional pressures on organizations to take measures to reduce haze pollution,
the relationship between high-speed rail and haze pollution vary under different conditions of the operational environment- ICT and market development. It is not clear why the authors
have only chosen to study these two mechanisms. There are other mechanisms which can also influence the pollution levels for e.g. the enforcement mechanisms of actions to protect environment by the
local governments. These lurking variables like monitoring imposed by the government, corruption, and others can also influence pollution . It is not clear how the authors control for these omitted variables.
The authors use instrumental variable to control for endogenous treatment. The instrument used is historical information on the operation of conventional train tracks. The authors suggest that the operation
of high-speed rail has a relationship with the operation of conventional train tracks; however, the operation of conventional train tracks does not have a direct relationship with haze pollution.
The instrument used is not convincing. One can argue that the operation of conventional train tracks also reduces the distance between local government and firms and thereby influence haze pollution by
improving information asymmetry.
How did the authors control for selection bias caused by the non-randomness of HSR opening?
Other particular comments:
Line 263-272: Not all components of the model are explained clearly. For example: if control is a set of covariates, is the corresponding coefficient a scalar or vector of coefficients? Is there any random effect?
What are the model assumptions? If the observations are not independent, what is the corresponding covariance structure?
Line 390 and Table 5: What is the corresponding linear model here? Specify the instrumental variable regression model and the main model after incorporating the effect. Table 5 is not clearly explained.
What are these variables? Perhaps it would be good to clearly write these models here. What is GMM? If we assume it is Generalized Mixed Model, questions is what is random effect there (and why?) so that authors had to
resort to GMM?
- In general, please provide the corresponding statistical model so readers can follow more clearly.
Author Response
Thanks for giving us the unique opportunity to revise our work. It means a lot to us.
In short, we mainly revised this piece of work in two ways. First, we strengthened our empirical work with an effort to handle the endogeneity issues (i.e., selection bias). Second, we improved the language by using the MDPI copyediting service. Specifically, one-by-one response are presented as follows.
Point1: The authors suggest for Hypothesis 1, that the "operation of high-speed rail has improved information transparency, which increases institutional pressure on organizations to reduce haze pollution." It is not clear what the authors' argument is in terms of making causal inference based on a model. The authors argue that although the operation of high-speed rail decreases information asymmetry and increases institutional pressures on organizations to take measures to reduce haze pollution, the relationship between high-speed rail and haze pollution vary under different conditions of the operational environment- ICT and market development. It is not clear why the authors have only chosen to study these two mechanisms. There are other mechanisms which can also influence the pollution levels for e.g. the enforcement mechanisms of actions to protect environment by the local governments. These lurking variables like monitoring imposed by the government, corruption, and others can also influence pollution. It is not clear how the authors control for these omitted variables.
Response1: Thanks for your seminal comments. For any study its mission should to establish the causal relationship. Thus we also want to build up this relationship in this study. However, we are very careful to use this word because as you pointed out we are subject to many drawbacks that may prevent us from pursuing that mission. To our best, in the current version, we have fixed this issue as follows. First, we included one more control variable to tease out the alternative explanation of government monitoring. Second, from an empirical aspect, we adopted the Heckman Two-Stage models to control for the selection bias. Third, as you saw, we introduced two moderating factors to strengthen our core logic. We think if our main argument, the institutional pressure is correct, then we should observe that this logic is working weak when the external market and ICT is relatively well developed because in these cases, the information asymmetry has been likely mitigated to a large extent before the HSR operation.
Based on these kinds of work, we think the relationship we established in the paper is more likely to go toward to the inference interpretation.
Point2: The authors use instrumental variable to control for endogenous treatment. The instrument used is historical information on the operation of conventional train tracks. The authors suggest that the operation of high-speed rail has a relationship with the operation of conventional train tracks; however, the operation of conventional train tracks does not have a direct relationship with haze pollution. The instrument used is not convincing. One can argue that the operation of conventional train tracks also reduces the distance between local government and firms and thereby influence haze pollution by improving information asymmetry. How did the authors control for selection bias caused by the non-randomness of HSR opening?
Response2: Yes, to be straightforward, we tried to use the HSR as a unique shock to study its effect on pollution or other issues as many scholars have done. As you pointed out, actually, the operation of HSR in one city may not perfectly exogenous. Thus, in our submitted version, we addressed this issue by using an instrumental variable, the year gap between the opening of conventional trains and HSR. Cities opened conventional trains have a higher possibility to open HSRs because most of Chinese HSRs were built based on conventional train lines. Yes, you are right, this is not strictly exogenous. In the current version, we address this point by using this IV as well as another one, the city latitude to construct a Heckman Two-Stage model to address this endogeneity issue (i.e., selection bias). The rationale for the second IV is that most of China’s HSRs were built at cities with lower latitude.
Point3: Line 263-272: Not all components of the model are explained clearly. For example: if control is a set of covariates, is the corresponding coefficient a scalar or vector of coefficients? Is there any random effect? What are the model assumptions? If the observations are not independent, what is the corresponding covariance structure?
Response3: Sorry for this ambiguity. For our baseline model, we utilized generalized estimating equations (GEEs). This method is appropriate for the purposes of this paper, as it addresses unobserved heterogeneity between cities and the autocorrelation that results from the repeated measurement of these cities over time. Thus, it thus combines the advantages of fixed and random models for panel data. We specified a log link function, a Gaussian family, and an independent correlation structure for all GEE models. In addition, we used robust standard errors to account for potential misspecifications of related structures and heteroscedasticity. In the current version, we clarified them.
Point4: Line 390 and Table 5: What is the corresponding linear model here? Specify the instrumental variable regression model and the main model after incorporating the effect. Table 5 is not clearly explained. What are these variables? Perhaps it would be good to clearly write these models here. What is GMM? If we assume it is Generalized Mixed Model, questions is what is random effect there (and why?) so that authors had to resort to GMM? - In general, please provide the corresponding statistical model so readers can follow more clearly.
Response4: Thanks for your valuable comment. Currently, we have chosen Heckman Two-Stage as the new method to address the endogenous problem attributed to selection bias. In the line 1155-1211, we have explained the linear model, the first stage and the second stage. In table 5 we have described all variables. In addition, the GMM is the Generalized Method of Moments.
Reviewer 3 Report
Please find the comments attached.

Author Response
Thanks for giving us the unique opportunity to revise our work. It means a lot to us.
In short, we mainly revised this piece of work in two ways. First, we strengthened our empirical work with an effort to handle the endogeneity issues (i.e., selection bias). Second, we improved the language by using the MDPI copyediting service. Specifically, one-by-one response are presented as follows.
Point1: The paper is poorly written. The language is awkward, and the paper has countless grammar errors. The authors didn’t use professional and scientific written language. The authors need to improve the language substantially.
Response1: We have used the MDPI language service to improve our language. In this way, we hope that the language issues have been largely solved.
Point2:The authors didn’t include necessary citations, which is unacceptable in scientific writings.
Response2: We have added concise citations in the right place, and deleted some unnecessary citations.
Point3: The authors didn’t used past tense when describing their work.
Response3: Sorry for this mistake. Now we have revised the present tense into past tense.
Point4: Numerous errors appear when the authors use some terms or words such as PM2.5, SO2, and PM10.
Response4: We have revised the form of SO2, CO2, NO2, PM10, PM2.5 in our paper.
Point5: Line 207-213. Move this paragraph before hypothesis 1.
Response5: This paragraph was used to connect the main hypothesis and the hypothesis of moderator variables. As you suggested, we modified it.
Point6: The 2nd and 3rd hypothesis are similar and can be united.
Response6: You made a great comment. In our original version, we have discussed this with coauthors. Finally, we decided to keep them as separate ones. The reasoning is twofold. One is that they present two different aspects of conditional factors, the hard and soft infrastructures. The second reason is that they are quantitatively different, and their correlation is 0.07, very small, indicating they are two different issues. We thus keep them, and believe they could provide more insights into our key logic underlying this study, the information transparency logic.
The detailed comments are listed as below.
1. Line 14. Substitute “whereas” with “but”. Change “fewer” to “few”.
Response: Yes, changed. Line 14
2. Line 16. Substitute “with” with “caused by” or some other terms.
Response: Yes, changed. Line17
3. Line 17-19. This sentence is not connected to the previous sentence very well.
Response: We have rephrased the sentences. Line18
4. Line 34. Change “particulate matter (PM) that can be inhaled” to “the inhalation of particulate matter (PM)”.
Response: Yes, changed. Line34
5. Line 37. Reformat the subscripts.
Response: Yes, we have reformatted all subscripts of the paper, such as SO2, CO2, NO2, PM10, PM2.5.
6. Line 38. Delete “thus.it has become a source of pollution of serious concern”.
Response: Yes. Line38.
7. Line 45. Substitute “and” with “as well as”.
Response: Yes. Line88
8. Line 47. Change “fewer” to “few”.
Response: Yes. Line90.
9. Line 50. Substitute “because of” with “in terms of”.
Response: Yes. Line93.
10. Line 52. indicates->indicated. Please check the tense of sentences all over the manuscript.
Response: Yes, we have revised all over the sentences about the tense.
11. Line 53. Change “and” to “,”.
Response: Yes. Line119.
12. Line 56. “Yang, Lin, Li and He [33]”->”Yang et al. [33]”. Change the format of references all over the manuscript.
Response: Yes, we have revised the reference format according to the SUSTAINABILITY requirements.
13. Line 57-58. “which was selected…has attracted more attention”. It should be a separate sentence.
Response: Yes, line 120-121.
- Line 61. Change it to “can have long-term implications on…”
Response: Yes, line124
15. Line 63. “are”-> “is”
Response: Yes, line125
16. Line 65. Do not use “etc” in scientific article.
Response: Yes, we have used “and so on” to replace “etc”.
17. Lien 67. What does “gained legitimacy” mean?
Response: Gain legitimacy means firms getting through a series of environmental protection of activities to gain the support and recognition of government and stakeholders. Accordingly, we updated it.
18. Line 69. Rephrase “in the context of …with haze pollution”.
Response: We have rephrased the sentence in Line 131.We wanted to express that “in the context of China”.
19. Line 73-75. Rephrase “when the market development…on the information environment”.
Response: Yes, we have used “undeveloped” to replace “backward”.
20. Line 81-83. Are the results from this study? If yes, put it into the results section. If not, the authors should include the citation here.
Response: This paragraph is the results of this study, now we have put it in the results section.
21. What is “a battery of mechanism tests”?
Response: We wanted to express a series of mechanism tests, and in the revision version we have deleted it.
22. Line 101. What is “aerodynamic equivalent”?
Response: We have revised the expression, we just wanted to express the “weather bureau”.Line156.
23. Line 101-102. “PM10, which can deposit on the upper respiratory system”.
Response: Yes, Line157.
24. Line 103. “directly”-> “deeply”.
Response: Yes, line159.
25. Line 104. “because of its influence on visibility”?
Response: We have revised it. “because of its visibility.”
26. Line 105. Rephrase.
Response: Yes, Line 161-162.
27. Line 109-110. Rephrase.
Response: Yes, line164-166.
28. Line 112. Check the tense. Please check the tense all over the manuscript.
Response: Yes, we have revised tense all over the manuscript.
29. Line 113. Use a separate sentence for the description of the study of Zhang et al.
Response: Yes, line336.
30. Line 128. Delete “To fill the research gap on haze pollution”
Response: Yes, line349.
31. Line 133. “other transportation”?
Response: Yes, we have revised it with “other modes of transportation”.
32. Line 135-140. Make this part concise.
Response: Yes, we have deleted the specific description section.Line358.
33. Line 142. early-> original?
Response: Yes, line360.
34. Line 142. What is “movement of individuals”? “Transportation”?
Response: Yes, we have revised the “movement of individuals” with “transportation”. Line360.
35. Line 143. Rephrase “such as gathering…accesibility”.
Response: Yes, we have rephrased this sentence. Line361-362.
36. Line 148. “Leinson….” should be in a separate sentence.
Response: Yes, we have revised it in line366.
37. Line 149. What “opportunities”?
Response: We have revised it, this sentence expresses that the operation of high-speed rail has provided opportunities for local cities to improve economic development.
38. Line 154. The authors used too much “argue” in this manuscript.
Response: We have increased other expressions, such as contend, hold the opinion, highlight, indicate.
39. Line 161. What does “serious influence” mean? What is “reputation damage”?
Response: We have revised with “bad influence” and “irresponsible image”.
40. Line 162-164. This sentence is not connected with the previous one well. Reorganize the praragraph.
Response: Yes, we have reorganized the sentence in line380-381.
41. Line 164. Include citation of “Global Burdern of Disease Assessment Report”. Also, the authors should have one or two sentences to illustrate how traffic emissions can influence public health. Here are some review papers.
1) Respiratory health effects of air pollution: Update on biomass smoke and traffic pollution, Robert J.LaumbachMD, MPHHoward M.KipenMD, MPH, Journal of Allergy and Clinical Immunology, Volume 129, Issue 1, January 2012, Pages 3-11
2) Traffic-related particulate matter and cardiometabolic syndrome: a reviewCM Ahmed, H Jiang, JY Chen, YH Lin, Atmosphere 9 (9), 336
3) A review of traffic-related air pollution exposure assessment studies in the developing world,X Han, LP Naeher - Environment international, 2006
Response: Yes, we have added some of these references in line381-382.
42. Line 168. “Considerable potential”? Correct it.
Response: Yes, we have deleted it.
43. Line 169. Rephrase.
Response: Yes, we have rephrased this sentence in line 386-389.
44. Line 173. Rephrase “because of…distance”.
Response: Yes, we have rephrased it in line605.
45. Line 175. “higher governmental levels”-> “higher levels of government”. Fix another place.
Response: Yes, we have revised it in all manuscript.
46. Line 178. “city”-> “cities”
Response: Yes, we have revised in line610.
47. Line 178. What does “paying attention to” mean?
Response: Yes, we have revised it with “focusing on”.
48. Line 182-184. Rephrase.
Response: Yes, we have revised it in line 614-616.
49. Line 186-187. Rephrase.
Response: Yes, we have rephrased it in line618-619.
50. Line 189. What does “make less demands on them” mean?
Response: we have revised it with “make less demands on the companies”.
51. Line 192. Correct the grammar error.
Response: Yes, we have used MDPI language edit service to correct it and others.
52. Line 201. Correct the grammar error.
Response: Yes. Line633.
53. Line 218. Delete “In his dissertation”.
Response: Yes. Line649.
54. Line 221. What does “increased for 27 out of 100 persons” mean?
Response: Yes, we have revised it with “increased by 27 out of 100 persons”.
55. Line 224. “limited influence”? Does is mean high-speed rails are not helpful in reducing haze pollution? It’s controversial to what the authors mentioned in other places.
Response: We have revised the “limited influence” with “lesser influence”.
56. Line 230. What does “closed” mean here?
Response: We have revised “closed” with “enclosed” in line723.
57. Line 268. “table 1”-> “Table 1”. Also fix other places.
Response: Yes, we have revised the format in manuscript.
58. Line 273. If “Information communication technology (Phonepeople)” is the title of this paragraph, format it different from other content.
Response: The bold format of “Information communication technology (Phonepeople)” is for highlighting.
59. Line 312. Do not use bold format for the title of Table. Reformat “HSR dummy….HSR=0”.
Response: We have revised it with “HSR dummy:If the city opens an HSR in the observation year, HSR = 1; otherwise, HSR = 0.”
60. Line 317. What is “our final sample”
Response: We have revised it with “our sample”.
61. Line 318. ug-> “μg”.
Response: Yes, we have revised it.
62. Line 328. If the results are from previous study, the authors should clearly mention it. It looks like the results are from this study, but the authors include a citation. It’s confusing.
Response: The results are from this study, we just cite this reference to express that the results are rational. In order to avoid confusing, we have deleted the reference.
63. Line 331. Table 4
Response: We have deleted the bold format.
64. Line 349. What is the title of y-aixs?
Response: The title of y-aixs is “haze surface concentration”.
65. Table 4. Is “-” the minus sign? The authors should reformat the table.
Response: Yes, we have revised it with “minus sign”.
66. Line 392. Are “previous results” from this study or other studies? It’s confusing.
Response: This paragraph we have went on major revision, and in the first version we wanted to express the endogenous result is consistent with the H1.
67. Line 399-400. Rephrase.
Response: This paragraph we have went on major revision and language editing service.
68. Table 7 seems to have larger font size than other tables.
Response: we have went on major revision about the robustness check section, so table7 has been deleted.
Reviewer 4 Report
The study assesses the effect of city-to-city high speed rail transportation on haze pollution. In particular, the development of hard and soft infrastructure is investigated through an analysis based on a sample of 288 case study cities in China over the period 2005-2016.
The work is perfectly in line with the scope of of Sustainability and the topic is interesting and a very live issue. That said, I have only two comments.
1) "Theoretical backgound" and "Data and sample sections". I suggest to better explain the main features of the mathematical model and in particular how the different control variables influence the environmental pollution indicator.
2) The authors focus their analysis on PM10 as representative of haze pollution level and the assessment is based only on the operation phase of the vehicles. That said, literature provides some papers which deal with the environmental impact of railway transportation mode by assessing a series of specific impact categories through the Life Cycle Assessment methodology. Such a type of studies is not limited to the use phase but it takes into account the entire life-cycle of the railway vehicles. Additionally, the inventory is based on a wide set of data regarding material and energy resources consumption as well as emissions to the environment provided by each one of life-cycle phases. In this regard the authors could expand their state of the art by mentioning some LCAs in the railway field. Additionally, the authors could evaluate the possibility to extand their analysis to other emissions or LCA indicator. Below I report some interesting papers that can be used for these purposes.
- Del Pero, F., Delogu, M., Pierini, M., Bonaffini, D., 2015. Life Cycle Assessment of a Heavy Metro Train, J Clean Prod, 87 (2015), pp. 787-799. https://doi.org/10.1016/j.jclepro.2014.10.023 - Chester, Horvath, 2010. Life-cycle assessment of high-speed rail: the case of California. Environmental Research Letters 5, 014003 - Delogu, M., Del Pero, F., Berzi, L., Pierini, M., Bonaffini, D., 2017. End-of-Life in the railway sector: Analysis of recyclability and recoverability for different vehicle case studies. Waste Management, Volume 60, February 2017, Pages 439-450. https://doi.org/10.1016/j.wasman.2016.09.034 - Chester, M.V., Horvath, A., 2009. Environmental assessment of passenger transportation should include infrastructure and supply chains. Environmental Research Letters 4, 024008.
Author Response
Thanks for giving us the unique opportunity to revise our work. It means a lot to us.
In short, we mainly revised this piece of work in two ways. First, we strengthened our empirical work with an effort to handle the endogeneity issues (i.e., selection bias). Second, we improved the language by using the MDPI copyediting service. Specifically, one-by-one response are presented as follows.
Point1: "Theoretical backgound" and "Data and sample sections". I suggest to better explain the main features of the mathematical model and in particular how the different control variables influence the environmental pollution indicator.
Response1: We have explained more clearly about the relation between control variables with haze pollution. In addition, we have explained more about the mathematical model in “Data and Sample sections” and “Endogenous Treatment”.
Point2: The authors focus their analysis on PM10 as representative of haze pollution level and the assessment is based only on the operation phase of the vehicles. That said, literature provides some papers which deal with the environmental impact of railway transportation mode by assessing a series of specific impact categories through the Life Cycle Assessment methodology. Such a type of studies is not limited to the use phase but it takes into account the entire life-cycle of the railway vehicles. Additionally, the inventory is based on a wide set of data regarding material and energy resources consumption as well as emissions to the environment provided by each one of life-cycle phases. In this regard the authors could expand their state of the art by mentioning some LCAs in the railway field.
Response2: Yes, you are right, the LCAs is good framework to construct and capture the different types of pollution. While, in this study, it seems go beyond our study. Here, we focus on a more visible form of pollution (PM10) that is in line with our theoretical foundation, the institutional pressure view. We acknowledge this is a valuable comment, and we thus consider it as a limitation of our study that deserves future study.
Point3: Additionally, the authors could evaluate the possibility to extand their analysis to other emissions or LCA indicator. Below I report some interesting papers that can be used for these purposes.
Response3: Thanks, although we have no detailed data about the LCA indicators, we think that enriching our analyses using other pollution indicators might provide more insights. To achieve this, in the robustness analyses, we added other two additional pollution indicators, i.e., SO2, CO2, the results are consistent with H1.
Round 2
Reviewer 1 Report
The editing still need to be checked.
Please adapt Fig.1 and 2 for the quality/format/look of images. The figures are deformed.
After, the paper can be accepted for publication.
Author Response
Thank you for your suggestion.
Point1: The editing still need to be checked.
Response1: We have edited the language again by using the MDPI language editing service.
Point2: Please adapt Fig.1 and 2 for the quality/format/look of images. The figures are deformed. After, the paper can be accepted for publication.
Response2: Thank you for your suggestion. We have edited all figures of the manuscript according to the format of published articles of SUSTAINABILITY journal. Line79, line81, line1942.
Reviewer 3 Report
Thank you for the authors to address the previous comments. I would appreciate it if the authors can refer to the line numbers correspondingly. It is hard for the reviewer to follow what the authors have modified. I still have a few minor comments. Figure 1. I don't think this figure is acceptable. It seems like that the figure is directly copied from a website. This is not acceptable. The authors have to redraw the figure on their own and include any needed references. Also this figure is
Figure 2. This figure is not suitable for a academic publication. The authors should improve it. Also, SO2 and some other symbol need to have subscripts. Figure 3. What is the y-axis? I don't see the authors include these citations as recommended previously.
Line 164. Include citation of “Global Burdern of Disease Assessment Report”. Also, the authors should have one or two sentences to illustrate how traffic emissions can influence public health. Here are some review papers.
1) Respiratory health effects of air pollution: Update on biomass smoke and traffic pollution, Robert J.LaumbachMD, MPHHoward M.KipenMD, MPH, Journal of Allergy and Clinical Immunology, Volume 129, Issue 1, January 2012, Pages 3-11
2) Traffic-related particulate matter and cardiometabolic syndrome: a reviewCM Ahmed, H Jiang, JY Chen, YH Lin, Atmosphere 9 (9), 336
3) A review of traffic-related air pollution exposure assessment studies in the developing world, X Han, LP Naeher - Environment international, 2006
Author Response
Thank you very much for your suggestion.
Point1: Figure 1. I don't think this figure is acceptable. It seems like that the figure is directly copied from a website. This is not acceptable. The authors have to redraw the figure on their own and include any needed references. Also this figure is Figure 2. This figure is not suitable for a academic publication. The authors should improve it. Also, SO2 and some other symbol need to have subscripts. Figure 3. What is the y-axis? I don't see the authors include these citations as recommended previously.
Response1: Thanks a lot for your suggestion. First, we have redrawn the figure1 according to the information of Chinese HSR opening. Second, we have edited the figure2 according to the published articles of SUSTAINABILITY journal. Third, we have revised the figure3 according to the study of Yang, Lin, Li, and He (2019). Line79, line81, line1942.
Point2: Line 164. Include citation of “Global Burdern of Disease Assessment Report”. Also, the authors should have one or two sentences to illustrate how traffic emissions can influence public health. Here are some review papers.
Response2: We have included the citation of “Global Burdern of Disease Assessment Report” (Bourne & Collaborators, 2016). In addition, we have illustrated the influence of traffic pollution on public health according to the study of Laumbach and Kipen (2012) and Ahmed, Jiang, Chen, and Lin (2018). Line437-line439.
Ahmed, C., Jiang, H., Chen, J. Y., & Lin, Y.-H. 2018. Traffic-related particulate matter and cardiometabolic syndrome: a review. Atmosphere, 9(9): 336.
Bourne, R. R., & Collaborators, G. R. F. 2016. Global, regional, and national comparative risk assessment of 79 behavioural, environmental and occupational, and metabolic risks or clusters of risks, 1990-2015: a systematic analysis for the Global Burden of Disease Study 2015. The Lancet, 388(10053): 1659-1724.
Laumbach, R. J., & Kipen, H. M. 2012. Respiratory health effects of air pollution: update on biomass smoke and traffic pollution. Journal of allergy and clinical immunology, 129(1): 3-11.
Yang, X., Lin, S., Li, Y., & He, M. 2019. Can high-speed rail reduce environmental pollution? Evidence from China. Journal of Cleaner Production, 239: 118135.